# Detecting Hallucinated Content in Conditional Neural Sequence Generation

## Abstract

Neural sequence models can generate highly fluent sentences but recent studies have also shown that they are also prone to hallucinate additional content not supported by the input, which can cause a lack of trust in the model. To better assess the faithfulness of the machine outputs, we propose a new task to predict whether each token in the output sequence is hallucinated conditioned on the source input, and collect new manually annotated evaluation sets for this task. We also introduce a novel method for learning to model hallucination detection, based on pretrained language models fine tuned on synthetic data that includes automatically inserted hallucinations. Experiments on machine translation and abstract text summarization demonstrate the effectiveness of our proposed approach – we obtain an average F1 of around 0.6 across all the benchmark datasets. Furthermore, we demonstrate how to use the token-level hallucination labels to define a fine-grained loss over the target sequence in the low-resource machine translation and achieve significant improvements over strong baseline methods. We will also release our annotated data and code for future research.

## 1 Introduction

Neural sequence models have achieved impressive breakthroughs in a wide range of applications, including data-to-text generation (Puduppully et al., 2019), machine translation (Vaswani et al., 2017; Wu et al., 2016) and text summarization (Rothe et al., 2020). Although these models can generate fluent sentences that are even sometimes *preferred* to human-written content (Läubli et al., 2018; Brown et al., 2020), recent work has also shown that they lack global logical consistency (Marcus & Davis, 2020), sometimes degenerate to dull and repetitive outputs (Welleck et al., 2019) and can often hallucinate content that is not entailed by the input (Maynez et al., 2020). In this paper, we focus on the faithfulness of machine outputs in conditional sequence generation tasks, aiming to automatically identify and quantify content in the output that is not faithful to the input text.

This risk of generating unfaithful content impedes the safe deployment of neural sequence generation models. The first step to building models that do not suffer from these failures is the assessment and identification of such hallucinated outputs. Prior work has shown that standard metrics used for sequence evaluation, such as BLEU scores (Papineni et al., 2002; Post, 2018), ROUGE (Lin & Hovy, 2004) and BERTScores (Zhang et al., 2019), do not correlate well with the faithfulness of model outputs (Maynez et al., 2020; Wang & Sennrich, 2020; Tian et al., 2019), and they also require reference text, limiting their applicability to detecting halluciations in a deployed system at run-time. Very recent efforts (Maynez et al., 2020; Durmus et al., 2020; Wang et al., 2020a) have started to develop automatic metrics to measure the faithfulness of output sequences. These methods use external semantic models, e.g. the question-generation and question-answering systems (Wang et al., 2020a; Durmus et al., 2020) or textual entailment inference models, to score faithfulness tailored for abstract text summarization. However, these scores do not directly measure the number of hallucinated tokens and only correlate weakly with human judgements due to compounded errors.

We propose a new task for faithfulness assessment - hallucination detection at the token level, which aims to predict if each token in the machine output is a hallucinated or faithful to the source input. This task does not use the reference output to assess faithfulness, which offers us the ability to apply it in the online generation scenario where references are not available. Similar to the spirit of our proposed task, word-level quality estimation (Fonseca et al., 2019) in the machine translation

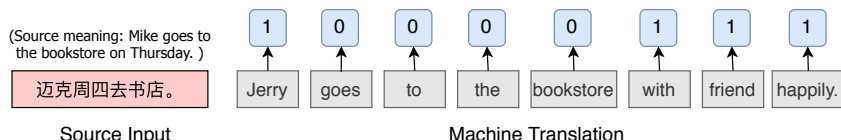

Figure 1: A toy example of token-level hallucination detection from machine translation. The words in grey blocks is an example of machine translation output and the labels above them indicate if each word is faithful (0) to the source input or a hallucinated one (1).

community predicts if tokens are correctly translated based on human post-editing. However, they do not distinguish errors in terms of fluency and adequacy (Specia et al., 2011). In contrast to estimating the amount of human post-editing work required to fix errors, we specifically focus only on hallucination (not fluency) errors.

We measure hallucination for two conditional sequence generation tasks – abstractive summarization and machine translation (MT). For the former, we produce a benchmark dataset from the recent released annotations in (Maynez et al., 2020). For MT, we carefully design the human assessment guideline and create high-quality annotations. We will also release our human annotated data for future research. To learn token-level hallucination prediction for general conditional sequence generations tasks, we propose a novel method that creates synthetic "hallucinated" dataset with pseudo labels and finetunes a pretrained language model (Liu et al., 2019; Conneau et al., 2020) on it. Without any human annotated supervised training data, we achieve an average F1 of around 0.6 across all the benchmark datasets, setting initial performance levels for this new task. We also show that pretraining on MT can actually produce more faithful translations, confirming recent findings in abstractive summarization (Maynez et al., 2020).

Predicting hallucination labels at token-level provides a tool for diagnosing and interpreting model outputs, which allows us to flag potential risks at inference time for previously unseen inputs. On the other hand, the token-level labels also offer possibility of fine-grained controls over the target sequence to improve the generation. We show how to use these token-level hallucination labels to improve self-training in low-resource MT, where the teacher can produce hallucinated outputs that are harmful to the student model. However, many outputs are only partially hallucinated (see examples in Appendix D.6) and the rest of the output is still useful for training, as we show by introducing different token-level loss truncation schemes. Our best method outperforms strong baselines by a large margin both in translation quality and hallucination reduction.

## 2 TASK: HALLUCINATION PREDICTION AT TOKEN-LEVEL

For a source sequence $S$ and its model generation $G$ from a neural conditional generation model, following Maynez et al. (2020) we define any span $w_i, \cdots, w_{i+j}(j >= 0)$ in $G$ as being hallucinated if it cannot be entailed by the source input $S$. More specifically, we consider two not mutually exclusive types of hallucination:

**Content Insertion:** a span $w_i, \cdots, w_{i+j}$ in $G$ consists of additional content that is not supported by $S$, i.e. its paraphrase or other equivalent form cannot be inferred from $S$. In Fig. 1, the word *"happily"* in the machine translation belongs to this case. This is also referred as "extrinsic hallucinations" in Maynez et al. (2020).

**Incorrect Substitution:** a span of word(s) is misrepresented information based on the $S$. In Fig. 1, *"Jerry"* in the machine translation is a hallucinated word and should be replaced by *"Mike"*. This type of hallucination is similar to the concept of "intrinsic hallucination" in Maynez et al. (2020). Note that there are cases where certain words (e.g. "This is not a great book." becomes "This is a great book.") are dropped in $G$ and hence the meaning of $S$ is changed, and we consider any spans in $G$ that misrepresent $S$ as hallucinated contents (e.g. the entire sentence of "This is a great book.").

We aim to identify all the span(s) satisfying the above conditions in the machine generation $G$.[1] Note that the above definition is only used in the guidelines for human annotators, who do not need to distinguish between these types rigorously.

---

[1] We do not consider the under-generations e.g. the source input is only partially translated or summarized.

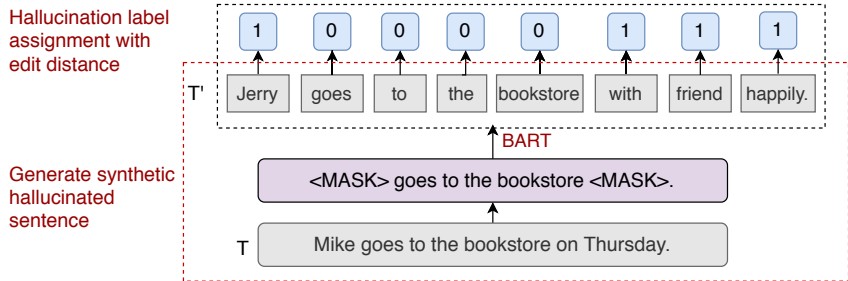

Figure 2: Generation of synthetic hallucinated data set with hallucination labels. The bottom block generates a hallucinated version of $T$ by feeding the noised sentence to the pretrained encoder-decoder model BART. The top block assigns hallucination labels to each token by computing the edit-distance between $T'$ and $T$. Labels of 1 refer to hallucinated words and vice versa.

## 3 LEARNING TOKEN-LEVEL HALLUCINATION DETECTION

We propose a general-purpose method for token-level hallucination detection for conditional sequence generation tasks. Given the source input $S$, we first formulate the task of token-level hallucination detection as a sequence labeling problem where a binary label is predicted at each position $G_t$ of the machine generation $G$. One straightforward way of learning such task is to train a model with supervised data in the form of $((S, G), L_G)$ where $L_G$ are the labels at every positions of $G$ that indicate if each word is a hallucinated one or not. However, because such labeled training data is not readily available, we also propose a approach to automatically create synthetic training data.

### 3.1 SYNTHETIC DATA CREATION

We use bi-text from the training data to create synthetic examples by automatically inserting new, hallucinated tokens. More speficially, we take target sequence $T$ and create a hallucinated version of it denoted $T'$ with associated hallucination labels for each token in $T'$. Then we can train a supervised model on this synthetic labeled data set of $((S, T'), L_{T'})$. The key challenge is that $T'$ should be a fluent sentence that does not differ too much from $T$.

**Generation of hallucinated sentences** To control this synthetic hallucination process, we build on recent work on large scale denoising autoencoders. These models learn to map a corrupted sentence back to the original text it was derived from, including for example learning to reconstruct missing words that have been arbitrarily masked out. We use BART (Lewis et al., 2020) by first inserting noises and then reconstructing parts of the target sentence, without providing it any access to the source sentence, thereby encouraging it to insert new content as needed to ensure fluency. As shown in Fig. 2, we first apply noising functions to the target sentence $T$ in the bi-text and then use a pretrained BART to generate $T'$ conditioned on the noised $T$ with beam search.

**Label assignments** After obtaining the hallucinated sentence $T'$ with BART, we need to assign appropriate labels to each token in $T'$ to mark which words are hallucinated. We compute the edit distance between $T'$ and $T$, and back-trace the deletion and substitution operations with dynamic programming. All the positions in $T'$ involving these two operations are labeled as hallucinations and everything else is

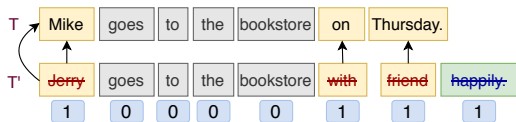

Figure 3: An example of label assignment.

considered faithful to $T$. Fig. 3 shows an example of label assignment with edit distance, where words in red are replaced and the word in blue is deleted to convert $T'$ to $T$. Assigning labels with edit-distance can not always guarantee the correct labels, however, we find that this simple approach provides sufficiently high quality training data for effective hallucination detection in practice.

### 3.2 FINETUNING PRETRAINED LM ON SYNTHETIC DATA

**Hallucination prediction loss** We follow the common practice in natural language understanding (NLU) tasks and finetune a pretrained language model (LM) on our synthetic data. We finetune a cross-lingual LM (Conneau et al., 2020) for machine translation and a monolingual model (Liu et al., 2019) for summarization. In both cases, we concatenate the input source, true target and hallucinated target denoted $(S, T, T')$ as a single input sequence to the pretrained LM. Then we

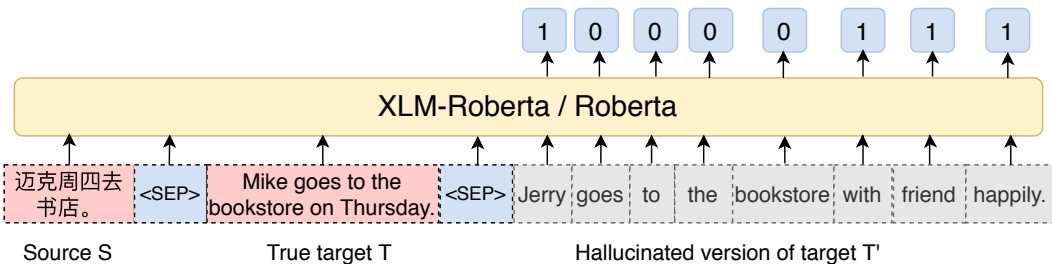

Figure 4: Finetuning XLM-Roberta (for cross-lingual generation task, e.g. machine translation) or Roberta (for monolingual generation task, e.g. text summarization) on the synthetic training data.

minimize the standard classification loss $\mathcal{L}_{pred}$ over the pseudo hallucination labels $L_{T'}$ on top of the final hidden vectors of each token in $T'$ as shown in Fig. 4.

Although using only the source text and hallucinated target $(S, T')$ as the input should be sufficient to learn the hallucination prediction task, we can also easily measure the extent to which including the true target $T$ in the input could help the model. At test time, when evaluating the faithfulness of the machine outputs $G$, we do not use the true target $T$ and perhaps surprisingly find our model can generalize well without references, even when it was present during training.

To prevent the model from overly relying on the true target $T$ and learning spurious correlations (e.g. the edit distance), we explored two techniques: (1) *dropout* – randomly drop out tokens in $T$ to force the dependence on the source input; (2) *paraphrase* – recall that at synthetic data generation time, we generate $T'$ from BART conditioned on the noised $T$. Instead, we can apply noise functions to the paraphrased sentence of $T$. Let $D$ denote the paraphrased sentence of $T$ and $D'$ denote the generation from BART conditioned on the noised $D$. Then we create pseudo labels of $D'$ denoted $L_{D'}$ by computing the edit-distance between the $D'$ and $D$ and use $((S, T, D'), L_{D'})$ as the training data for finetuning. Since the pseudo labels are created based on $D$, it can prevent the model from learning the edit-distance between $T$ and $D'$ easily. Ablation studies will be provided in § 5.4.

**Masked LM loss** We also add the masked language model loss (MLM) $\mathcal{L}_{mlm}$ following (Devlin et al., 2019). To learn this loss, we create a different batch from the above by concatenating only the source $S$ and target $T$ as the input, since the hallucinated target $T'$ could provide erroneous information for predicting masked words in $T$. We find that such multi-task learning objective helps learn better representations of the inputs and further improves performance on predicting hallucination labels. The final loss is $\mathcal{L} = \mathcal{L}_{pred} + \alpha \cdot \mathcal{L}_{mlm}$ where $\alpha$ is a hyperparameter.

## 4 EVALUATION TASKS AND DATA

We evaluate hallucination in conditional neural sequence models on abstractive text summarization and machine translation (MT) tasks, using the models and datasets described below.

### 4.1 ABSTRACTIVE TEXT SUMMARIZATION

Maynez et al. (2020) studied the hallucination problems in extreme summarization on the XSUM dataset which comprises 226,711 British Broadcasting Corporation (BBC) articles paired with their single-sentence summaries. They randomly sampled 500 articles from the test set of XSUM and evaluated summaries from four abstractive summarization systems: (1) **PtGen** (See et al., 2017), an RNN-based attentional sequence-to-sequence (Seq2Seq) model. (2) **TConvS2S** (Narayan et al., 2018) is a topic-aware convolutional (Gehring et al., 2017) Seq2Seq model. (3) **TranS2S** and (4) **BERTS2S** are standard Transformer (Vaswani et al., 2017) encoder-decoder models. The weights in TranS2S are randomly initialized while the encoder and decoder of BERTS2S are initialized with the pretrained BERT-base (Devlin et al., 2019) checkpoint. Maynez et al. (2020) asked the human annotators to label the spans in the machine generated summaries if they were unfaithful to the article (for more details see Maynez et al. (2020)). We post-processed their human annotations by majority voting and create our test datasets for each of the abstractive summarization systems.

### 4.2 MACHINE TRANSLATION

Previous work (Wang & Sennrich, 2020; Müller et al., 2019; Koehn & Knowles, 2017) has shown that translation models are particularly prone to hallucination when tested out of domain. We simi-

larly focus on this evaluation regime and additionally consider the low resource case where a modest amount of out of domain data is available at training time.

**Data** We use a multi-domain Chinese-English (Zh-En) translation dataset (Wang et al., 2020b) which consists of four balanced domains: *law, news, patent* and *subtitles*. We create a new training data $\mathcal{D}_{train}$ with *law* (1.46M sentences), *news* (1.54M), *subtitles* (1.77M) train data and randomly sample 0.03% parallel sentences from the *patent* (870) training data. We train two NMT models (described below) on this dataset and test on 150 examples sampled from patent test data. In addition, we also evaluate the trained models on the COVID-19 domain. We sample 100 examples from the recently released translation benchmark dataset (Anastasopoulos et al., 2020). Together with the 150 examples from patent test data, we create our benchmark dataset with 250 examples in total and we denote this dataset $\mathcal{D}_{eval}$. We ask human annotators to evaluate the level of hallucinations on this dataset $\mathcal{D}_{eval}$.

**Models** Our evaluation data is generated from two reference models on which we will measure hallucination (see Appendix for more details): (1) **TranS2S** (Vaswani et al., 2017) is the standard Transformer Seq2Seq model with 6 encoder layers and 6 decoder layers. (2) **MBART** (Liu et al., 2020) is a Seq2Seq denoising auto-encoder pretrained on large-scale monolingual corpora in many languages. We finetune a MBART checkpoint on $\mathcal{D}_{train}$ which is a standard Transformer architecture with 12 layers of encoder and 12 layers of decoder pretrained on a multilingual corpora of 100 languages.

| Models | Fleiss' Kappa | |
| --- | --- | --- |
| | Token | Sent |
| MT | | |
| TranS2S | 0.58 | 0.72 |
| MBART | 0.54 | 0.62 |
| XSum | | |
| PtGen | 0.81 | - |
| TConvS2S | 0.83 | - |
| TranS2S | 0.79 | - |
| BERTS2S | 0.79 | - |

Table 1: Fleiss's Kappa scores (↑): agreements on **token**-level hallucination labels or sentence-level (**sent**) ratings among different annotators. The token-level agreements for XSUM are computed on the released annotations by Maynez et al. (2020).

### 4.3 HUMAN ASSESSMENT OF HALLUCINATIONS

In the human evaluation, three bilingual annotators were presented the Chinese source sentence, the English reference sentence and the machine translation. We conducted the pilot study and practice sessions with annotators before annotating the blind test set $\mathcal{D}_{eval}$. Annotators were asked to only focus on the hallucinations during assessment and label each sentence with one of the three types of labels: incomprehensible, faithful, and hallucinated. If the translation is a hallucinated one, we asked the annotators to tag all the tokens that were not faithful to the source. We dropped all the translations that were labeled as incomprehensible (15 for TranS2S and 3 for MBART). The final benchmark datasets were created by taking majority labels among three annotators. (See Appendix A for more annotation details.)

In Tab. 1, we show the Fleiss's Kappa scores of our annotations for MT and the processed annotations from (Maynez et al., 2020) on abstractive summarization. A higher Fleiss's Kappa score indicates higher agreements among annotators.[2] We achieved moderate agreement on the token-level hallucination annotations and substantial agreement on the sentence-level annotations, while Maynez et al. (2020) achieved substantial or almost perfect agreement on the XSUM dataset. For MT, it's relatively harder to achieve consistent agreements among annotators for the following reasons: first, although we have made strict annotation guidelines following the definition of hallucination in Section 2, it could still be difficult for annotators to distinguish between bad translations and hallucination; second, it was sometimes difficult for annotations to understand the specialized text in the patent domain, which can contain complex scientific descriptions.

## 5 EXPERIMENTS

### 5.1 EXPERIMENTAL SETUP

**Synthetic Data Generation** We use a pretrained BART (Lewis et al., 2020) model in the fairseq toolkit (Ott et al., 2019) with 12 layers of encoder and 12 layers of decoder for synthetic labeled data generation. We uniformly sample the percentage of tokens $p_m$ to mask from $[0, h_m]$ for each sentence. We also uniformly sample the probability of replacing a token with a random token from $[0, h_r]$ denoted $p_r$. $p_m$ and $p_r$ are two important factors that affect the noise level when generating

---

[2]https://en.wikipedia.org/wiki/Fleiss%27_kappa

| Methods | MT | | Summarization | | | |
|---|---|---|---|---|---|---|
| | TranS2S | MBART | PtGen | TConvS2S | TranS2S | BERTS2S |
| Alignment | 29.47 | 9.93 | 38.92 | 37.94 | 34.47 | 35.81 |
| Overlap-based | 9.14 | 3.24 | 57.22 | 54.25 | 53.79 | 55.13 |
| Synonym-based | – | – | 59.54 | 63.73 | 58.66 | 53.07 |
| Ours (w/o reference) | **65.75** | **41.92** | 63.66 | 65.94 | 61.70 | 55.45 |
| Ours (w/o reference + synonym) | – | – | **64.72** | **69.37** | **63.88** | **56.49** |
| Ours (w/ reference) | 66.08 | 46.81 | 63.89 | 66.28 | 62.24 | 55.88 |

Table 2: F1 (x100) of hallucination labels on the outputs of different systems from a machine translation task (see§4.2) and the abstract summarization task (XSUM dataset). The first block are baseline methods and the second block are our results. We highlight the best results without using reference.

the synthetic data. For MT, we set $h_m$ and $h_r$ to 0.6 and 0.3 respectively. For abstractive summarization, we use 0.4 and 0.2. We use beam search for decoding from BART with beam size of 4 and length penalty of 3. For MT, we first create paraphrased target sentences $D'$ through knowledge distillation (Kim & Rush, 2016) by using the outputs from the same trained TranS2S model on the source inputs.

**Finetuning Pretrained LM**  For MT, we finetune XLM-R (Conneau et al., 2020) on the synthetic dataset with batch size of 128, and we annotated 50 examples (different from those in $\mathcal{D}_{eval}$) from the patent test data as the validation dataset. For summarization, we finetune RoBERTa (Liu et al., 2019) with batch size of 96 and early stop training with 10K update steps. In addition, we dropout tokens from the reference $T$ in the input with a rate of 0.5 and 0.3 respectively for summarization and MT to learn $\mathcal{L}_{pred}$. We set $\alpha$ to be 0.6 for MT and 0.5 for summarization based on the scales of $\mathcal{L}_{pred}$ and $\mathcal{L}_{mlm}$. For both tasks, we set the mask probability used for $\mathcal{L}_{mlm}$ to be 0.5, and the initial learning rate to be $2e-5$ with polynomial decay. We describe other hyperparameters, including training of MT models, in the Appendix B.

## 5.2 EVALUATION ON TOKEN-LEVEL HALLUCINATION PREDICTION

In Tab. 2, we present the F1 of token-level hallucination labels across six benchmark datasets for MT and astractive summarization (the full results of precision, recall and F1 are presented in Tab. 9 and Tab. 10 in appendix). We compare with three baseline methods that we proposed for this new task: (1) The **alignment-based** method uses a word alignment model for hallucination assessment. We employ SimAlign (Sabet et al., 2020), an unsupervised word aligner, that extracts word alignments from similarity matrices induced from pretrained word embeddings. SimAlign is essentially used for crosslingual tasks, and we adapt it to summarization by using embeddings from the pretrained BERT-large (Devlin et al., 2019). We predict a target token as being hallucinated if it is not aligned to the source tokens. (2) The **overlap-based** method is a heuristic one that predicts a target token as being hallucinated if does not appear in the source input. Since it's not feasible to perform string matching between two languages for MT, we use the bilingual lexicon induction method (Zhou et al., 2019) to first translate each English word into the Chinese word and then check its existence in the source text. (3) We go further by exploiting **synonyms** to assess hallucination in the summarization task where we use WordNet (Miller, 1998) to find synonyms of nouns, verbs, adjectives and adverbs of the target summary and the source article; we predict a target as being hallucinated if its synonym can not be found in the set of the source synonyms.

From Tab. 2, we have the following observations: (1) We achieve descent performance on this task and rank the best among all the baseline methods, however it is still far from being solved. Token-level hallucination prediction requires deep semantic understanding of the inputs and is worthy of study in the future. (2) We can see that even though our model learns hallucination prediction with reference $T$ during training (Sec. 3.2), by applying token dropout to $T$, our model generalizes well without feeding the reference at test time. As a contrast, we report the results of predicting with reference at test time and observe that the model can achieve a significantly higher recall but worse precision (Tab. 10). (3) The two non-neural baselines we proposed work surprisingly well on the summarization datasets, especially the synonym-based system. We guess this is because the information of the summaries should come from the source article and a majority of hallucinated words are nouns 5.3 which can be easily detected by string matching or synonym matching. Our neural system performs better than these baseline methods but not significantly, and we hypothesize

| Methods | MT | | Summarization | | | |
|---|---|---|---|---|---|---|
| | TranS2S | MBART | PtGen | TConvS2S | TranS2S | BERTS2S |
| True hal. tokens (%) | 18.12 | 11.10 | 46.09 | 52.89 | 46.74 | 37.51 |
| Pred hal. tokens (%) | 18.56 | 7.99 | 57.22 | 57.68 | 55.78 | 48.84 |

Table 3: Comparisons of annotated (True) and predicted (Pred) percentage of hallucinated tokens on the benchmark test sets.

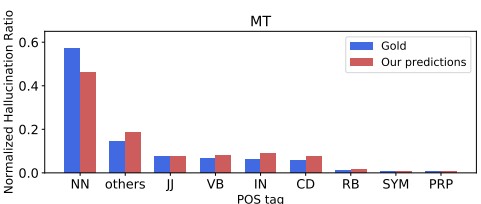
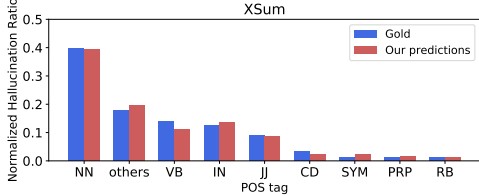

Figure 5: Relationship of Part-of-Speech tags and percentage of hallucinations for machine translation (left) and summarization (right).

that this is because the Roberta model we finetune on only allows a maximum input length of 512, which results in an average cutoff of 158 subwords from the source article and losses of the source information. By taking the union of the predictions from the synonym-based and our models, we can further obtain improvements on the summarization datasets. We believe the advances in long sequence modeling (Beltagy et al., 2020; Kitaev et al., 2020) are advantageous to our models. (4) At the same time, the baseline methods can not obtain reasonable performance for MT since crosslingual semantic matching is more challenging and our model shows significant improvements over these brittle baselines.

In Tab. 3, we show the percentage of annotated and our model predicted percentage of hallucinated tokens across the six benchmark test sets. We can see that model predictions correlate well with human assessment and has a Pearson correlation coefficient of 0.986.

## 5.3 ANALYSIS

**Analysis on Pretrained Models for Conditional Sequence Generation**   Recent work (Maynez et al., 2020) has shown that pretrained models are better at generating faithful summaries as evaluated by humans. In Tab. 3, summaries generated from BERTS2S contain significantly less hallucinations than other model outputs. We also confirmed this trend in machine translation that translations from MBART contain much less hallucinated contents than that from TranS2S.

**Analysis on Hallucination Words and their Part-of-Speech Tags**   In Fig. 5, we present the percentage of hallucinated tokens categorized by their Part-of-Speech tags predicted by a POS tagger (Toutanova et al., 2003). First, we see that for both MT and summarization datasets, nouns are the mostly hallucinated words. In abstractive summarization, verbs also account for certain amounts of hallucinations. Second, our model predicted hallucinated words match well with gold annotations on the distributions of POS tags. We also compare the percentage of hallucinations within each POS tag in Appendix §D.4.

## 5.4 ABLATION STUDIES

**Effects of including reference at training time**   Recall that we concatenate the source, reference and machine generation together as the input when learning hallucination predictions (Sec. 3.2). In Fig.6, we vary the dropout rate of tokens in the reference at training time and evaluate the models on the outputs from the TranS2S model for both tasks, where dropout rate of 1.0 indicates that we do not include the reference at all. **First**, different dropout rates do not signficinaly affect performance for

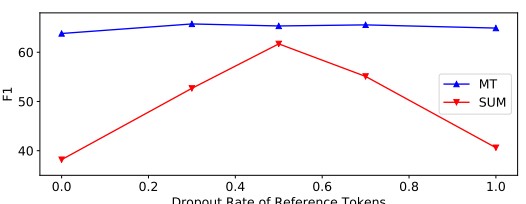

Figure 6: Performance on the TranS2S benchmark from MT and summarization by varying the dropout rate of tokens in the reference at learning hallucination predictions.

MT, this is likely because we use the paraphrased target when creating the synthetic data instead of the reference sentences. Thus, the "hallucinated" sentences $D'$ from BART do not resemble the reference $T$ as closely as $T'$, and the model will not learn spurious correlations between the $T$ and $D'$. **Second**, for summarization we see that applying word dropout is crucial since we have used the reference more directly for generating synthetic data. On the other hand, if reference is removed at learning time (dropout = 1.0), the resulted model performs poorly, which shows that including reference at training time also has positive effects.

**Effects of paraphrased data**   We investigate the effects of using paraphrased data in Tab. 4, where we apply the noise functions to different forms of targets when generating synthetic data. For MT, we create paraphrased targets via knowledge distillation (Kim & Rush, 2016) where we use the output from TranS2S conditioned on the source sentence in the bi-text corpus as the paraphrased target. We can see that with distillation data for synthetic data generation, the model achieves better results compared to using the references. However, note that we need to choose a proper word dropout rate when using the reference-based synthetic data as discussed above.

| Input to $\mathcal{N}(\cdot)$ | Precision | Recall | F1 |
|---|---|---|---|
| MT | | | |
| raw | 58.35 | 70.12 | 63.70 |
| TranS2S distill | 64.27 | 67.30 | 65.75 |
| Summarization | | | |
| raw | 57.02 | 67.23 | 61.70 |
| Extractive distill | 54.10 | 36.45 | 43.55 |
| Abstractive distill | 57.33 | 28.59 | 38.16 |

Table 4: Performance on the TranS2S benchmark from MT and summarization by using different data as the input to the noised function $\mathcal{N}(\cdot)$. "raw" refers to the original targets in the training data.

For abstractive summarization, we create paraphrased data out of an abstractive and an extractive summarization systems respectively. We finetune BART on the bi-text of XSUM and create distillation data from this finetuned abstractive model. For the extractive system, we use the recent proposed MatchSum (Zhong et al., 2020) as the distillation model. We see a significant drop in the performance for both of the variants. This likely due to the fact that: (1) it has been shown that abstractive summarization systems are prone to hallucinate contents themselves (Maynez et al., 2020), thus we are not able to create reliable pseudo labels based on the generated summaries, and (2) the extractive system generates summaries out of the input article which diverge from the actual abstractive summaries we evaluate on, and the model cannot generalize well under such data shift.

## 6   A CASE STUDY: IMPROVING SELF-SUPERVISED MACHINE TRANSLATION

Predicting hallucination labels at token-level not only allows us to flag potential risks in generation models, but also offers potential to improve text generation by defending against hallucinated outputs. Token-level hallucination labels provide fine-grained signals which can be used to define new learning objectives. In this section, we show such fine-grained signals allow for an improved semi-supervised method for machine translation.

### 6.1   RECTIFIED SELF-TRAINING IN LOW-RESOURCE NEURAL MACHINE TRANSLATION

Self-training (Scudder, 1965) is an important semi-supervised approach that utilizes unlabeled source data to improve system performance. In a conditional sequence generation task, a teacher model is first trained with bitext $\mathcal{D}_l = \{\mathbf{s}_i, \mathbf{t}_i\}_{i=1}^N$ and used to make predictions on each sequence in a unlabeled dataset $\mathcal{D}_u = \{\mathbf{s}_j\}_{j=N+1}^{N+M}$ to create *pseudo parallel data* $\mathcal{D}_p = \{\mathbf{s}_j, \mathbf{t}'_j\}_{j=N+1}^{N+M}$. The model is then trained on $\mathcal{D}_l \cup \mathcal{D}_p$. He et al. (2020) finds that with self-training the student model can benefit from such pseudo parallel data that acts as a regularizer. However, such results require a relatively high quality teacher, and performance suffers in low-resource setting where there isn't enough parallel data. In these cases, the teacher may hallucinate content when it makes predictions, thus producing noisy parallel data that actually hurts the student model.

We propose to use our token-level hallucination predictions as a fine-grained control during training in machine translation, by penalizing errors less on tokens that more likely to be hallucinated. This is in strong contrast to previous data filtering in MT, which showed it was possible to improve performance by removing entire sentence pairs (Junczys-Dowmunt, 2018; Kang & Hashimoto, 2020).

First, we predict the token-level hallucination labels on the target side of the pseudo parallel data $\mathcal{D}_p$. Then we propose two simple methods of using these labels in self-training: (1) We discard the losses of tokens that are predicted as hallucinations and compute the loss on the remaining tokens for

each target sequence (**token loss truncation**). (2) Instead of adjusting losses, we mask the decoder hidden states of those hallucinated positions after the target-to-source cross attention in each decoder layer (**decoder HS masking**).[3]

| Methods | BLEU (↑) | BLERUT (↑) | Hal words (%, ↓) |
| --- | --- | --- | --- |
| baseline | 16.14 | -0.166 | 13.69 |
| ST | 19.31 | -0.059 | 10.00 |
| ST + paraphrase noise | 19.05 | -0.051 | 13.48 |
| ST + random noise | 19.97 | -0.041 | 12.55 |
| ST + seq loss truncation | 19.91 | -0.048 | 8.26 |
| ST + random noise + seq loss truncation | 19.37 | -0.057 | 10.06 |
| ST + token loss truncation | 20.32 | 0.00244 | **6.37** |
| ST + decoder HS masking | 20.57 | -0.0001 | 6.38 |
| ST + random noise + token loss truncation | **21.02** | **0.043** | 7.34 |
| ST + random noise + decoder HS masking | 20.64 | 0.0308 | 8.70 |

Table 5: BLEU, BLEURT and percentage of hallucinated tokens on the CWMT2017 patent test set. We compare with the noised self-training method (He et al., 2020) in the second block and sequence-level loss truncation method (Kang & Hashimoto, 2020) in the third block.

## 6.2 EXPERIMENT SETUPS AND RESULTS

**Experimental Setup**  To train a teacher model (baseline in Tab. 5), we use the same training data described in §4.2 using *patent* (870) as the low-resource domain. We evaluate on the full patent test set (1,500) from CWMT2017 (Wang et al., 2020b). For the unlabeled data, we use the withheld Chinese patent training data (2.9M).

**Baselines**  First, we compare with the state-of-the-art self-training (ST) method (He et al., 2020) that injects noise to the inputs. They use two types of noise: (1) Paraphrase noise created by round-trip translations, and (2) Random noise from droping, masking and shuffling input tokens. We also compare with the recently proposed loss truncation method (Kang & Hashimoto, 2020) that adaptively removes entire examples with high log loss, which can reduce hallucinations as shown in the paper.

**Results and Analysis**  We present the tokenized BLEU score (Papineni et al., 2002), BLEURT score (Sellam et al., 2020) and the percentage of hallucinated tokens predicted by our system in Tab. 5. We can see that ST improves over the baseline by around 3 BLEU scores and our best result further improves ST by 1.7 BLEU scores. Compared with strong baseline methods, our method not only achieves the best translation quality measured by BLEU and BLEURT but also the largest hallucination reduction. We also observe that: (1) Our method with ST alone can outperform other baseline methods, when combined with perturbed ST (noise), using fine-grained control over the target tokens can further improve the results. (2) ST with paraphrase noise (by round-trip translations) does not perform as well as the random noise, which further confirms that the noisy outputs from a teacher model may hurt the student model. (3) The sequence-level loss truncation approach can improve over the vanilla ST and reduce level of hallucinations as measured by our system. However, the performance drops when combined with the noised ST.

## 7 CONCLUSIONS

In this work, we proposed a new evaluation task for hallucination detection in conditional sequence generation and created human-annotated benchmark datasets. We also proposed a novel and general-purpose method to learn this task. In the future, we hope to create a large-scale pretrained evaluation model for any datasets or models to be evaluated. We also would like to extend our method to any data-to-text generation scenarios. We are also interested in investigating how to leverage our detection methods to mitigate hallucination problems in conditional sequence generation.

---

[3]We also tried removing the hallucinated target words before training. This does not perform as well as the above methods, likely because it produces too many ungrammatical target sentences.

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

## A    HUMAN EVALUATIONS

We asked three bilingual speakers to annotate the Chinese-to-English evaluation set $\mathcal{D}_{eval}$, which is composed of 150 sentences from the test set of Zh-En multi-domain dataset (Wang et al., 2020b), and 100 sentences from the COVID-19 translation benchmark dataset (Anastasopoulos et al., 2020) – TICO. TICO contains 6 finegrained domains including *Wikisource, Wikivoyage, Wikinews, CMU, PubMed* and *Wikipedia*. we randomly sample 25 examples from each of the four domains – Wikisource, Wikinews, CMU and PubMed, use these 100 samples for evaluation. We train standard base Transformer Seq2Seq and finetune the MBART (Liu et al., 2020) model on the training data from $\mathcal{D}_{train}$.

**Annotation Guidelines**    We ask each annotator to read the tokens in the sentence carefully and check if they can be supported by the source sentence:

- If there are tokens (or the entire sentence) that cannot be supported by the source, label all the span(s) with color and mark the sentence as a hallucinated one;
- If the evaluator can not understand the entire translation at all, mark the sentence as incomprehensible;
- If all the tokens in the translation can be entailed from the source, mark the sentence as a faithful one.

We shuffled the orders of sentences so that annotators did not know which translation model was used (TranS2S or MBART). Before the annotation work on the final blind test set, we first perform a pilot study on a different evaluation set and analyze the annotation results from each annotators. Then we conduct an education session to make sure that all the evaluators can fully follow the guidelines.

Note that we ask annotators to work on the raw sentences, i.e. punctuation marks can also be labeled as hallucinations along with the span and we didn't apply special treatments to them. And all the evaluations are compared against the raw form of sentences, and we always convert the model predictions on subwords to labels on the raw sentences. Besides, based on our guidelines, annotators labeled the entire span of hallucinated words that may also contain prepositions and other stop words.

For each token in the evaluation set, we label it by majority voting, i.e. the label that two or more annotators agree on. We also aggregate the evaluation data from Maynez et al. (2020) in the same way to produce our own test set for abstract text summarization.

## B    TRAINING OF NMT MODELS: TRANS2S AND MBART

| Models | *TranS2S* | *MBART* |
|---|---|---|
| $d_{\text{model}}$ | 512 | 1024 |
| $d_{\text{hidden}}$ | 1024 | 4096 |
| $n_{\text{layers}}$ | 6 | 12 |
| $n_{\text{heads}}$ | 8 | 16 |
| $p_{\text{dropout}}$ | 0.1 | 0.3 |

Table 6: Basic hyper-parameters of architecture for NMT models.

**Tokenization**    For TranS2S, we first segment the Chinese corpus with an opensource Chinese word segmentation tool (Luo et al., 2019), then we learn separate BPE vocabularies with 32k merge operations (Sennrich et al., 2016) over the source (Zh) and the tokenized target (En) corpus respectively. For MBART, we directly apply the contained sentence-piece dictionary in the finetuned model to the raw data of Chinese and English corpus.

**Model**    We use the implementation of Transformer model from fairseq (Ott et al., 2019). Following the notations in Vaswani et al. (2017), we show hyperparameters of TranS2S and MBART in Tab. 6.

**Training**    For TranS2S, we apply the standard hyperparameters reported in the example of fairseq. We use the Adam optimizer (Kingma & Ba, 2014) using $\beta_1 = 0.9, \beta_2 = 0.98, \epsilon = 1e - 8$. The learning rate is scheduled using `inverse_sqrt` with a maximum learning rate $0.0005$ and $4000$ warmup steps. We set the label smoothing as $0.1$. We apply dropout of $0.1$ and select the best model with validation BLEU scores. We run the model on 8 GPUs for $300,000$ updates with an effective batch size of around $64,000$ tokens.

When finetuning MBART, we use learning rate of 3e-5, and use `polynomial_decay` for learning rate scheduling with warmup updates of 3,000. The effective batch size is 16,384. Dropout is set to be 0.3 and the attention dropout rate is 0.1. The label smoothing is set to be 0.2. We finetune MBart for 60,000 updates.

**Decoding**    After training, we use beam-search with a fixed beam size $5$ for all the models and set the maximum generation length to be twice as the source length.

## C    EXPERIMENTAL DETAILS FOR TOKEN-LEVEL HALLUCINATION PREDICTION

**Subword Tokenization**    Depending on the pretrained model (Roberta / XLM-Roberta) we finetune on, we apply corresponding subword segmentation to the synthetic data set $(S, T, T')$ and calculate the edit-distance between the $T$ and $T'$ at the subword level. At evaluation time, the model predicts the hallucination labels for each subword in the sentence, thus we predict a word to be a hallucination word if any subword of it is predicted as a hallucinated one.

**Synthetic data generation**    There are a couple of hyperparameters of noised functions in the BART implementation (Ott et al., 2019). The main noised functions include (1) random masking (randomly replace a word with MASK), (2) random replacement (randomly replace a word with another word in the vocabulary), (3) random insertion of masks. We found that random masking and random replacement are the two key factors affecting the generated sentences and we have provided their settings in the main paper. We apply a random insertion masks rate of 0.2 for all settings. In addition, the noise functions are applied to words instead of spans in our setting.

**Finetuning**    For MT, we finetune a large XLM-Roberta (Conneau et al., 2020) released in fairseq (Ott et al., 2019) which is trained on 2.5T of filtered CommonCrawl data in 100 languages. For summarization, we finetune a large Roberta (Ott et al., 2019) on the synthetic data where we truncate articles that exceed 512 tokens (allowed by the Roberta) to be 512. For both models, we use the Adam optimizer (Kingma & Ba, 2014) with $\beta_1 = 0.9, \beta_2 = 0.98, \epsilon = 1e - 6$ and weight decay of 0.1. We set the masking probability to be 0.35 for the $\mathcal{L}_{mlm}$ loss. The dropout and attention dropout rates are set to be 0.1. We adopt `polynomial_decay` for learning rate scheduling with learning rate of 2e-5.

## D    RESULTS AND ANALYSIS

### D.1    HALLUCINATION STATISTICS ON PATENT AND COVID-19

| Models | Patent | | COVID-19 | |
|---|---|---|---|---|
| | True hal. toks (%) | Pred hal. toks (%) | True hal. toks (%) | Pred hal. toks (%) |
| TranS2S | 22.78 | 22.83 | 9.44 | 11.79 |
| MBART | 8.84 | 6.94 | 6.39 | 5.55 |

Table 7: Comparisons of TranS2S and MBART on Patent and COVID-19 domains. **True hal. toks** and **Pred hal. toks** are the ground-truth and our model predicted percentage of hallucinated tokens in the benchmark test set for each domain.

We present the percentage of hallucinations from the annotations and our model predictions for the Patent and COVID-10 evaluation data respectively in Tab. 7. We see that both the TranS2S and

MBART models produce less hallucinations for the COVID-19 domain although it is a complete out-of-domain test set that has not seen in the training data. We suppose that this is because sentences in COVID-19 are more alike the training domains in terms of writing styles which bring positive transfer. We also note that the predictions from our model match pretty well with human annotations for both of the domains.

## D.2 COMPARISONS ON SENTENCE-LEVEL HALLUCINATION SCORES

|  | Entail | Align | Ours (P) | Ours (R) | Gold (%) |
|---|---|---|---|---|---|
| MT |  |  |  |  |  |
| TranS2S | -0.32 | 0.23 | 0.78 | **0.79** | 64.1 |
| MBART | -0.19 | 0.05 | 0.36 | **0.48** | 74.3 |
| Summarization |  |  |  |  |  |
| PtGen | 0.25 | 0.32 | **0.43** | 0.42 | 89.4 |
| TConvS2S | 0.26 | 0.31 | **0.43** | 0.42 | 92.8 |
| TranS2S | 0.13 | 0.19 | **0.35** | 0.33 | 92.4 |
| BERTS2S | 0.23 | 0.27 | **0.37** | 0.36 | 88.2 |

Table 8: Spearman's correlation coefficients of the sentence level hallucination scores computed by different systems with the human annotations (percentage of hallucinated tokens in a sentence). **Gold (%)** is the percentage of sentences that contain hallucinations in the test set.

We aggregate the token-level predictions to score the level of hallucination of each sentence and compute the Spearman's correlation coefficients between our scores and the human annotations of percentage of hallucinated tokens in a sentence. In Tab. 8, we present two scores from our model: (1) Probability-based score (**P**) computes the score by averaging the halluciation probabilities across all the tokes in $G$. (2) Ratio-based score (**R**) first predicts the hard labels by taking $\arg\max$ at each position and then use the percentage of hallucinated labels as the score.

We compare with two baseline metrics that do not require reference as well: (1) Textual entailment prediction model (**Entail**) has been proposed to evaluate the faithfulness of abstractive summaries (Maynez et al., 2020; Kryściński et al., 2019; Falke et al., 2019). We adapt it to MT by finetuning XLM-Roberta on the machine translated Zh-En Multi-NLI dataset (Conneau et al., 2018). For summarization, we use the finetuned Roberta-large (Ott et al., 2019) on the Multi-NLI dataset (Williams et al., 2018). We calculate the entailment probability $P_e$ between the source input and the machine output $G$ and use $1 - P_e$ to score its hallucination level (which is better than use of contradiction probability in our experiments). (2) For the alignment-based method (§5.2), we compute the percentage of tokens in the machine output $G$ that get aligned aligned to the source input to measure the faithfulness of $G$.

Overall, in Tab. 8 we see that: (1) For MT, we achieve very significant improvements over the baseline metrics. And both entailment and alignment scores correlate very poorly with the hallucination assessments. (2) For abstractive summarization, our metrics also significantly outperform baseline metrics. The alignment scores stand out as a better hallucination assessment compared to the commonly adopted entailment scores. We also note that for MT our ratio-based scores correlate better with human judgements while for abstractive summarization the probability-based scores have a better correlation. This shows that our model can produce more confident predictions for MT.

## D.3 FULL RESULTS OF TOKEN-LEVEL HALLUCINATION PREDICTIONS

We found the synonym and string-matching based methods are strong and effective baselines on the monolingual (summarization) token-level hallucination prediction task as an alternative to neural methods. However, previous work (Maynez et al., 2020; Wang et al., 2020a; Durmus et al., 2020) on hallucination assess did not study synonym-based non-neural baselines when measuring the faithfulness of the summarization model outputs.

| Methods | TranS2S | MBART |
|---|---|---|
| Alignment | (18.90, 66.82, 29.47) | (5.63, 42.09, 9.93) |
| Overlap-based | (7.02, 13.10, 9.14) | (1.98, 8.97, 3.24) |
| Synonym-based | – | – |
| Ours (w/o reference) | **(64.27, 67.30, 65.75)** | **(49.56, 36.32, 41.92)** |
| Ours (w/ refenrence) | (59.92, 74.27, 66.08) | (43.13, 53.63, 46.81) |

Table 9: Triplets represent (Precision, Recall, F1 (x100)) of hallucination labels on the outputs of different systems from a machine translation task (see§4.2). The first block are baseline methods and the second block are our results. We highlight the best results without using reference.

| Methods | PtGen | TConvS2S | TranS2S | BERTS2S |
|---|---|---|---|---|
| Alignment | (60.66, 28.65, 38.92) | (66.14, 26.60, 37.94) | (56.24, 24.85, 34.47) | (50.68, 27.69, 35.81) |
| Overlap-based | (67.72, 49.54, 57.22) | (60.39, 49.24, 54.25) | (53.22, 54.37, 53.79) | (62.57, 49.26, 55.13) |
| Synonym-based | (50.52, 72.50, 59.55) | (57.06, 72.16, 63.73) | (50.29, 70.37, 58.66) | (41.80, 72.67, 53.07) |
| Ours (w/o ref) | (57.47, 71.35, 63.66) | (63.21, 68.93, 65.94) | (57.02, 67.23, 61.70) | (49.83, 62.50, 55.45) |
| Ours (w/o ref + syn) | **(50.33, 90.27, 64.72)** | **(56.86, 88.93, 69.37** | **(50.21, 87.78, 63.88)** | **(41.70, 87.52, 56.49**) |
| Ours (w/ ref) | (56.51, 73.48, 63.89) | (61.68, 71.63, 66.28) | (55.88, 70.19, 62.24) | (48.39, 66.11, 55.88) |

Table 10: Triplets represent (Precision, Recall, F1 (x100)) of hallucination labels on the abstract summarization task (XSUM dataset). The first block are baseline methods and the second block are our results. We highlight the best results without using reference.

## D.4    ANALYSIS ON PART-OF-SPEECH TAGS AND WITH-IN GROUP HALLUCINATION PERCENTAGE

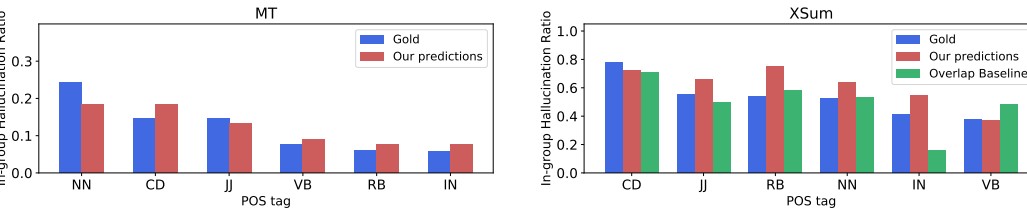

Figure 7: Analysis of part-of-speech tags and with-in-group percentage of hallucinations for machine translation (left) and summarization (right).

We have shown that the macro Part-of-Speech tag distribution of hallucinated tokens in §5.3. In this section, we analyze the micro-percentage of hallucination labels within each POS tags. We show the gold annotations as well as our model predictions of hallucination words within each POS tags. For summarization, we also show the results from the string-matching baseline. From Fig. 7, we can see that for MT nouns are most likely hallucinated words while for summarization cardinal numbers (e.g. *one, two*) are most likely hallucinated words. And we can see that our model predictions align well with the gold annotations on the percentage of hallucinated words within each POS tags.

## D.5    ANALYSIS ON SYNONYM PREDICTION ERRORS

In Tab. 11, we analyzed the error rates of hallucination predictions of the summarization data sets w.r.t. the synonyms of source words. Specifically, we first obtain the synonyms of nouns, adjectives, adverbs and verbs through WordNet (Miller, 1998), then for each word that belongs one of the mentioned POS tags in the model output we also obtain their synonym set. If the synonym set of a target word has overlap with the synonym set in the source article, we check if model predicts this word as a hallucinated word (if so, it's an error). In this way, we can analyze how well our model performs on synonyms, and we hope that our model can correctly identify synonyms as non-hallucinated words. However, we found that our model performs worse on synonym predictions

| Methods | Noun Synonyms | All Synonyms |
|---|---|---|
| String-match baseline | 15.37 | 17.40 |
| Ours (w/o reference) | 26.36 | 23.22 |
| Ours (w/o reference, filter overlapped NN with source) | 10.05 | 13.62 |

Table 11: Error rates of hallucination predictions on target words that are synonyms (predicted by WordNet) of words in the input on XSUM test sets.

than the string-matching baseline. As we pointed out in §5.2, the model we finetune on only permits a maximum input length of 512 subwords and many subwords of the source article are cut off at prediction time, which might be one reason that our model fails to identify the synonyms due to losses of source information. We integrate the string-matching based method at prediction time in a conservative manner where we always assign non-hallucination labels to *noun* target tokens that appear in the source. As shown in Tab. 11 (3rd row), we can reduce the error rates of synonym predictions by a large margin. Although we observe an increased precision, the integration of string-matching also leads to decreased recall and the F1 is not improved.

## D.6 EXAMPLES OF PARTIALLY HALLUCINATED OUTPUTS FROM TEACHER MT MODEL

| | |
|---|---|
| **Source** | 信息组被称作页面数据。 |
| **Reference** | the set of information is called page data. |
| **Generation** | the foreign[1] mix[1] is called the page data. |
| **Source** | 金属线对应于第一电阻器。 |
| **Reference** | the metal lines correspond to first resistors. |
| **Generation** | the wire corresponds with the first capital[1]. |
| **Source** | 平均液滴尺寸低于100nm。 |
| **Reference** | the average droplet sizes are below 100 nm. |
| **Generation** | the average exhaust[1] dimensions of the droplets were less than 100 cubic[1] meters[1]. |
| **Source** | 驱动样本流过液流通路; |
| **Reference** | driving samples to flow through a flow channel; |
| **Generation** | driving samples pass the flow of people[1]; |

Table 12: Examples of partially hallucinated outputs from the teacher MT model used in self-training and the hallucinated labels predicted by our system. We only highlight words with hallucination labels with [1].

In Tab. 12, we randomly select some examples for which we present the source sentences from the patent monolingual Chinese dataset, the corresponding reference English sentences and the generations from a teacher model trained on the training data described in §4.2 where patent is a low-resource domain. We can see that in these examples, only parts of the model outputs are hallucinated and the rest of the outputs are good translations that are faithful to the source. Through our approach in §6, we can still make use of these good parts of translation during training.

## D.7 EXAMPLES OF HALLUCINATION PREDICTIONS ON THE MT TEST SET

In Tab. 13, we present some examples of annotations and our model predictions. We can see that our model performs well in general but can be inaccurate in case of spelling errors of the translations. Besides, we also find some annotation errors while our model predicts correctly.

| Reference | the arrangement pattern of the projections 2 will now be explained with reference to figs. 5-7. |
|---|---|
| Annotation | next,[0] we[0] use[0] fig.[0] 5[0] -[0] 7[0] to[0] explain[0] the[0] disposition[0] pattern[0] with[0] pm-2.[1] |
| Prediction | next,[0] we[0] use[0] fig.[0] 5[0] -[0] 7[0] to[0] explain[0] the[0] disposition[0] pattern[0] with[1] pm-2.[1] |
| Reference | a swivel joint 557 is provided in a radially outer region, on an end surface of the drive plate 556. |
| Annotation | a[0] rotation[0] hinged[1] 557[0] is[0] provided[0] to[0] the[0] external[0] area[0] on[0] a[0] trail[1] that[0] has[0] a[0] preface[1] state.[1] |
| Prediction | a[0] rotation[0] hinged[0] 557[0] is[0] provided[1] to[0] the[0] external[0] area[0] on[0] a[0] trail[1] that[1] has[1] a[0] preface[1] state.[1] |
| Reference | r is small, the kinetic energy e of the droplet is small as compared to the air resistance. |
| Annotation | on[0] the[0] other[0] hand,[0] radius[0] r,[0] which[0] is[0] shorter[1] than[1] the[0] hour[1] in[0] which[1] it[1] can[1] be[1] used,[1] is[0] smaller[0] than[0] the[0] air[0] resistance.[0] |
| Prediction | on[0] the[0] other[0] hand,[0] radius[0] r,[0] which[0] is[0] shorter[1] than[0] the[0] hour[0] in[1] which[1] it[1] can[1] be[1] used,[1] is[0] smaller[0] than[0] the[0] air[0] resistance.[0] |
| Reference | if you have a fever of a hundred and two or higher. |
| Annotation | if[0] your[0] heat[0] reaches[0] 102.d[0] egree.[0] f.[0] or[0] above,[0] |
| Prediction | if[0] your[0] heat[0] reaches[0] 102.d[1] egree.[1] f.[1] or[0] above,[0] |

Table 13: Examples of annotations and model predictions. [0] indicates faithful word while [1] indicates hallucinated word.

