# OpenReview forum: "Detecting Hallucinated Content in Conditional Neural Sequence Generation"
_ICLR.cc/2021/Conference — Reject_

### Official Review · AnonReviewer1 · 2020-10-24
**new task**

**Rating:** 5
**Confidence:** 4

**Review:**

This paper proposes a new task: Word-level hallucination detection in text generation, where the authors identified two scenarios of word-level hallucination (content insertion and incorrect substitution).

Then the author synthesized hallucinating text by de-noisng auto encoder and use dynamic programming to infer hallucination labels. They are used to fine-tune pertained LMs for detecting hallucination.

Experiments show that the fine-tuned LMs can achieve certain performance compared with human annotations, and that the token-level hallucination is correlated to sentence-level hallucination.

Concerns:

1) Hallucination is recognized as an important issue for text generation, and intuitively, detecting hallucination is also an important task. But the real benefit of detecting hallucination is unclear from this paper. I would be more convinced if the authors could show how their word-level hallucination can indeed help text generation.

Currently, the paper appears to be some fine-tuned LM for a new task (whose real benefit is unclear) with little technical depth. At this stage, I am unsure if this paper has enough contribution as an ICLR paper.

2) The categorization of "content insertion" and "incorrect substitution" is unclear and confusing.

Is there any breakdown analysis on "content insertion" and "incorrect substitution"? For the below example, what type is "Monday" given Input1 or Input2, respectively?

Input1: We had a meeting in that room.

Input2: We had a meeting.

Gen: We had a meeting Monday.

3) There's very little technical development and empirical analysis on the role of synonyms and stop words in hallucination.

A synonym is supposed to have the same meaning as the original word, and is not hallucination. Synonyms are actually desired in many applications, like paraphrase generation and summarization.

Stop words mainly serve for grammatical functions, and may not be directly related to content hallucination.

---

> ### Author Response · Authors · 2020-11-18
> **Thank you for the feedback! We addressed your concerns.**
>
> Thank you and we really appreciate your valuable feedback! Please see our response below:
>
> > (1) The real benefits of predicting hallucination at word-level.
>
> Please look at our general response (1), where we highlight two main benefits of this new task, in particular we demonstrate how to use token-level hallucination labels to improve low-resource machine translation in Section 6 of our revised paper. We also list other potential ways of using these fine-grained labels to improve text generation in our general response (1)(b).
>
> > (2) The definition of hallucination is confusing.
>
> Thank you for pointing this out! In fact, the two categorizations are defined from the perspective of edit-distance. Given this, for the example you gave, “Monday” is replacement word for input 1 and inserted word for input 2. However, we also stressed in the revised paper that the definition of hallucination is only used in the guidelines for human annotators, who do not need to rigorously distinguish between these categories.
>
> > (3) Lack of analysis / experiments on synonyms and stop words
>
> Thank you for the great suggestion!  For the synonyms, we first propose to use them as a baseline method for the token-level hallucination prediction task in Section 5.2 (Table. 2) for summarization. Specifically, we examine if the synonym of a target word appears in the synonym set of source articles (if not, we predict it as being hallucinated). And this is a strong baseline to our method while previous work has not explored it for assessment of hallucinations. In Section 5.2, we also analyzed the reason why our model only slightly outperforms the baseline methods by a small margin for summarization, basically, the Roberta model we fine-tune on only permits a maximum input length of 512 subwords, which causes an average cutoff of 158 subwords from the source article. We believe advances in long-sequence modeling (e.g. Reformer) could further improve our method. The above is reflected in our revisions as well.
>
> Second, we analyzed the error rates of synonyms in Appendix D.5, where we found our method has a higher error rate on predicting synonyms as hallucinated words. However, when combined with the noun-matching technique, we achieve much lower error rate and high precision. But we also observe the drop in recall, and the F1 is not improved.
>
> For stop words, we would like to clarify that annotators are asked to label the entire span of hallucinated parts, which may contain prepositions, stop words. And stop words sometimes are also important for meaning expressions, e.g. "She is a student."  v.s. "She is not a student". With "not", the latter sentence is completely hallucinations compared to the former one. In Fig. 5 and Fig. 7, we also analyze the POS tags on hallucination labels and found certain amounts of stop words (e.g. prepositions) are marked as hallucinations.

---

### Official Review · AnonReviewer4 · 2020-10-27
**Official blind review**

**Rating:** 5
**Confidence:** 4

**Review:**


Summary:
This paper proposes hallucination detection at the token level, which predicts if each token in the generation output is hallucinated or faithful to the source input. In contrast, previous studies usually work on the sentence level. To create synthetic training data, a denoising pre-trained LM is first used to generate (potentially) unfaithful counterparts T’ of the references T. Then, token-level labels are obtained by comparing T and T’ via edit distance. Finally, a standard classification model is trained on the token-level labels by concatenating the source S, true and unfaithful targets T (T’).

---
Pros:
+ Overall, the paper is clearly presented with detailed experiments and ablation studies. It shows that the proposed method works better than existing entailment-based and alignment-based methods on the new task that it proposes.
+ The new annotated data on token-level hallucination could possibly be useful for future research in similar directions.

---
Cons:
- My major concern is about the effectiveness of defining hallucination at the token level itself. First of all, it looks un-intuitive to me whether it makes sense to define hallucination for every type of token. As the task aims to label every token in the text sequence, I wonder how different types of tokens (e.g., tokens with different POS tags) contribute to the model hallucination. The proposed metrics treat all the tokens equally, while in reality, tokens such as noun phrases or verb, for example, may have a larger impact on the hallucination issue than prepositions or articles. Similarly, for the creation of synthetic training data, I wonder whether it makes sense to replace tokens with different POS tags uniformly. It will be good to present more analysis in terms of both quantitative results and case studies on this aspect.
- Is the evaluation on all the tokens or only the hallucinated tokens? Could the decent performance (F1) of the proposed method come from the fact that most tokens are not hallucinated (labeled with 0)?
- For comparisons on sentence-level hallucination (of abstractive summarization), why don’t you compare with the baselines presented in the related papers you listed directly?

Typo: "if each token in the machine output is **a** hallucinated or faithful to the source input"

---
Comments after reading author response and revised paper

Thanks for showing more results of how token-level hallucination detection can be useful/effective: e.g., (i) labels per POS tagging in Fig. 5 and (ii) application to low-resource MT in Sec. 6.

For (i), I don't think it resolved my question directly: "The proposed metrics treat all the tokens equally, while in reality, tokens such as noun phrases or verb, for example, may have a larger impact on the hallucination issue than prepositions or articles." That is, a hallucinated NN, for example, might be worse than a hallucinated II, rather than asking how the labels are distributed by POS categories. So I still wonder if it makes sense (as a reliable metric) to measure hallucination at the token level (e.g., Table 5 Hal words %) but it remains unanswered.

For (ii), it is indeed an interesting plus to the paper, showing that the token-level labels appear to be useful for downstream applications (even if it's not quite meaningful to measure the % of hallucinated words). I would suggest doing more studies on downstream tasks as mentioned in your response to enhance the paper if you are "not proposing a reference-free evaluation metric for quality estimation" (which seems a bit contradictory to "we hope to create a large-scale pretrained evaluation
model for any datasets or models to be evaluated" in the conclusion section btw).

---

> ### Author Response · Authors · 2020-11-18
> **Thank you for the feedback! We addressed your concerns.**
>
> Thank you and we really appreciate your valuable feedback! Please see our response below:
>
> > (1) Questions on defining token-level hallucinations and lack of analysis on POS tags.
>
> Thank you for the great suggestions! First, please look at our general response (1),  where we highlight two main benefits of this new task, in particular we demonstrate how to use token-level hallucination labels to improve low-resource machine translation in Section 6 of our revised paper. We also list other potential ways of using these fine-grained labels to improve text generation in the general response (1)(b).
>
> Second, we added analysis experiments of POS tags in Section 5.3 and Appendix D.4 of our revised paper. Basically, we found that nouns / verbs / adjectives indeed accounts for a large portion of hallucinated words, however, we also found that certain amounts of stop words (e.g. prepositions) are marked as hallucinations both in the gold annotations and in our model predictions. And we would like to clarify that annotators were asked to label the entire span of hallucinated parts, which may contain prepositions and stop words. In addition, stop words sometimes are also important for meaning expressions, e.g. "She is a student." v.s. "She is not a student". With "not",  the latter sentence is completely hallucinations to the former one.
>
> From our analysis, we also found that the distribution of POS tags of our model predictions of hallucinated words align well with that of the gold annotations.
>
> > (2) Is the evaluation on all the tokens or only the hallucinated tokens?
>
> We measure F1 only on hallucinated tokens.
>
> > (3) Baseline of sentence-level hallucination evaluation
>
> Thanks for your question! The only baseline we listed and we didn’t compare with is the question-answering system which uses a question generation system to generate questions on the model output and use a QA system to answer questions based on document and summary respectively. In the original paper of the proposed method, they present a low correlation with human judgements (<0.2). In (Maynez et al., 2020), they have evaluated this baseline on XSum, and show a correlation of only 0.044.
> As the sentence-level hallucination scoring is not the main contribution of our paper, we moved this section to Appendix. In addition, we added three baseline methods to the token-level hallucination prediction (which is our task of interest) as well as more complete analysis in Section 5.2 of the revised version.

---

### Official Review · AnonReviewer2 · 2020-10-27
**Official Blind Review #2**

**Rating:** 6
**Confidence:** 4

**Review:**

SUMMARY

This paper presents a method to detect hallucinated tokens in generations from neural machine translation and summarization. Given a source input S and its output G generated by a sequence generation model, this paper formalizes the task of detecting hallucinated tokens as a labeling problem on the output G. In order to train the labeler, the method synthetically generates supervision data by using a BART model. The BART model receives a text with noises ([MASK] tokens) and tries to predict [MASK] tokens. In this way, the method obtains a pseudo hallucinated text T' from a text T, and assigns hallucination labels by estimating edit operations between T and T'. The labeler is trained by fine-tuning pre-trained cross-lingual (for MT) and mono-lingual (for summarization) language models. In training, the labeler receives a source text S, true target text T, and pseudo hallucinated text T' separated by [SEP] tokens and tries to reproduce the hallucination labels on T'. Receiving a source text S and its output G, the labeler predicts hallucination labels on G during the inference time.

The experiments use the XSUM dataset for summarization and multi-domain Chinese-English (Zh-En) translation dataset for machine translation. The authors manually prepared test sets for evaluating the hallucination labeler. In Section 6.2, the authors reported that they achieved decent performance on this task of hallucination labeling although it is far from being solved. They also aggregated token-level hallucination predictions to sentence-level scoring and computed the Spearman's rank correlation coefficients between the scores and the percentage of tokens annotated as hallucinations by humans. The comparison between the proposed method and two baselines (entail and align) show that the proposed scores correlate well with human judges than the baselines.

PROS

This paper is clearly written and easy to read.

The methods used in this study, e.g., BART for generating pseudo hallucinated text and (XLM-)RoBERTa for labeling hallucinated tokens, are appropriate.

CONS

Findings from the experiments are unclear. In addition, this paper does not explicitly explain how the research outcomes contribute to an advance in MT or summarization. In particular, I'm not sure whether the performance of Table 2 was successful enough to advance the research in MT and summarization. The goal of the correlation analysis in Table 3 is unclear. I guess that the initial motivation for this experiment was to develop an automatic method for evaluating hallucinations, but the current experimental results do not support this so strongly.

QUESTIONS

Please report the number of hallucinated and faithful sentences in the test set. This paper explains that the test set for machine translation included 250 instances initially, but removed instances judged as incomprehensible. I have no idea how many translations were hallucinated and faithful. This may be important because the Entail baseline works not at token level but at sentence level (it should not be used on a dataset only with non-entailment instances).

I would like to see results of a simple baseline in Table 2. For example, we can consider a baseline that indicates hallucinations for tokens that do not appear in the source text. Currently, I'm not sure how good the performance values of Table 1 are. Hence, I have no idea whether the method for generating pseudo supervision data and for detecting hallucinations at token level was a worthy contribution.

I am wondering of the usefulness of the evaluation with Spearman's rank correlation coefficients. Here is a summary of the methods.

Reference values: the percentage of tokens annotated as hallucinations by humans.

Proposed model (1): the average of hallucination probabilities across all the tokens.

Proposed model (2): the percentage of tokens labeled as hallucinations by the model.

Entail: the probability estimate of entailment relation between the source and output.

Align: the percentage of tokens aligned to the source input estimated by SimAlign.

The Entail baseline was not designed to work at token level but at sentence level. There is no guarantee that a probability estimate from the baseline represents the 'degree' of entailment relation that corresponds to the percentage of tokens. The comparison would be fairer if the proposed model could yield a binary decision about hallucination/faithful at sentence level.

In addition, Spearman's rank correlation coefficients of the Entail baseline on the MT dataset were mostly negative (weak inverse correlation). If -0.32 could be obtained by chance, we can conclude that coefficients around 0.3-0.4 do not indicate a correlation. I also suspect that this baseline could not work well because most instances on the MT dataset were non-entailment instances.

P3: Figure 2

Does the BART model often produce multiple tokens (e.g., "with friend happily") from a single [MASK] token? This example is impressive, but I think that a BERT model mostly predicts a single token for a single [MASK] token.

P3: "we consider any spans in G that misrepresent S as hallucinated contents"

This treatment may cause an inconsistency in labeling hallucinated tokens. How does this treatment affect the evaluations in Section 5?

P4: It was difficult to follow the description of "paraphrase". How does this study generate paraphrases?

MINOR COMMENTS

P7: "aligned aligned" -> "aligned"

COMMENTS AFTER THE REVISION

It was amazing to see the authors updated paper with the new experiment on the baselines (Table 2) and machine translation (Section 6). Although the baselines (overlap and synonym) were strong, I can now see that the proposed approach is better than these simple baselines both for machine translation and summarization. Section 6 also demonstrates the usefulness of this work for machine translation trained with self-training. For this reason, I increased my rating.

The impact of this paper would be greater if Section 6 could include more results on different MT datasets (e.g., WMT) and/or self-training for summarization.

---

> ### Author Response · Authors · 2020-11-18
> **Thank you for the feedback! We addressed your concerns.**
>
> Thank you and we really appreciate your valuable feedback! Please see our response below:
>
> > (1) How does predicting hallucination at word-level contribute to MT or summarization?
>
> Please look at our general response (1), where we highlight two main benefits of this new task, in particular we demonstrate how to use token-level hallucination labels to improve low-resource machine translation in Section 6 of our revised paper. We also list other potential ways of using these fine-grained labels to improve text generation in our general response (1)(b).
>
> > (2) Findings from the experiments are unclear and Question 2 - lack of simple baselines in Table 2.
>
> We apologize that our previous presentation of Table 2 is unclear and lacks interpretability. Please see our general response (2) to this question. In our revised version, we added three baseline methods which include the string-matching as you suggested (for MT, we leverage bilingual lexicon induction to first translate the target word into the source language). In addition, we added more complete analysis to the experimental results in Section 5.2. We found the string-matching and synonym-matching methods are strong baselines for the summarization task and analyzed why our neural model only outperform non-neural baselines by a small margin. Basically, the Roberta model we finetune only permits a maximum input length of 512 subwords, which causes an average cutoff of 158 subwords from the source article. We believe advances in long-sequence modeling (e.g. Reformer) could further improve our method.
>
> > (3) Please report the number of hallucinated and faithful sentences in the test set.
>
> Thanks for your suggestions! In our revised version, we report the percentage of hallucinated sentences in Table 8 in Appendix D.2, hallucinated tokens in Table 3. and number of incomprehensible sentences in Section 4.3 (15 for TranS2S and 3 for MBART).
>
> > (4) Questions on the correlation analysis in Table 3.
>
> We apologize that the results of sentence-level hallucination scoring can be distracting in our main paper, in our revised version we moved  this section to Appendix D.2. We agree with you that the entailment baseline might not be proper to reflect the percentage of hallucinated tokens and is not an appropriate baseline for the MT task, either. We would like to clarify that the reason we use it as a baseline method is to make comparisons with previous work on sentence-level hallucination assessment, where entailment method is shown to be one approach that works well to some extent. However, our focus is token-level hallucination detection, thus we added more baseline methods and analysis to this task in Section 5.2.
>
> > (5) Question on if the BART model often produces multiple tokens.
>
> Yes, it does often produce multiple tokens. BART uses an autoregressive decoder to generate output, instead of the mask-filling tasks performed by BERT.
>
> > (6) How does the definition of hallucination affect the evaluation?
>
> Thank you for pointing this out! We stressed in the revised paper that the definition of hallucination is only used in the guidelines for human annotators, who do not need to rigorously distinguish between the two categorizations. The model predictions are only compared with human annotations at evaluation time for the aggregate prediction (hallucination vs non-hallucination).
>
> > (7) What does “paraphrase” do?
>
> We apologize this is not clear from our paper. We generate “paraphrase” by first training a source-to-target base model using the bi-text, then feeding the source sentences in the bi-text to this base model and using the outputs from the base model as “paraphrases”.

---

### Official Review · AnonReviewer3 · 2020-11-02
**Interesting approach to identify hallucination, but have doubts about its use as an evaluation metric**

**Rating:** 5
**Confidence:** 4

**Review:**

Summary: The paper addresses the problem of "hallucinated" content in conditional neural generation for two specific tasks: machine translation and summarization. It proposes a new task for faithfulness assessment, which classifies each token as either hallucinated or not. The classifier uses a pre-trained LM (either XLM-R or ROBERTa) and is fine-tuned on synthetic classification data created using both 'noisified' real data and a pretrained LM (BART). Experiments on either summarization and MT system outputs labeled for hallucinations show relatively encouraging classification results (e.g., F1 of 0.46 to 0.66 for MT, and 0.56 to 0.66 for summarization).

Pros:

Addressing the hallucination problem is essential, as inserting non-factual content is probably the most harmful error a machine translation or summarization system can make.

The paper contributes a new dataset for training classifiers for hallucination detection and a methodology for creating new ones relatively easily. That said, the dataset is partially synthetic and may not reflect the kind of hallucinations actual neural MT and summarization systems make.

The paper contributes an evaluation task for assessing the faithfulness of MT and summarization output. It could be used, e.g., to flag or prevent output whose content is deemed inaccurate. The paper shows an improved correlation (Spearman) relative to an entailment metric and a word alignment metric.

Cons:

1. Lack of experimental comparison: I find it hard to conclude much from the paper's main results, as it doesn't provide a comparison across different hallucination detection systems or models. In general, it is difficult to tell if a given absolute score (e.g., F1 = 0.60 mentioned in the introduction) is good or bad without a point of comparison, especially if the task is new and there is relatively little information about the type of content errors the underlying systems make. I understand that the authors are trying to establish a new task and that there is little work to compare against, but the paper could have provided more in the way of ablation experiments.

2. Results lack interpretability: Two of the main tables (Table 2 and 4) do not seem to contribute much, as the different rows correspond to different datasets (underlying data to classify is different) and are therefore strictly not comparable. For example, Tab. 2 suggests that the classifier does a much better job on TransS2S (F1 = 0.66) than on MBART (F1 = 0.47). However, Tab. 5 indicates that MBART is much less prone to producing hallucinated words than TransS2S, which probably explains why classification results on MBART data are worse. So it seems Tab. 2 and 5 together suggest classification results are lower when the classification task is harder, which wouldn't be a surprising finding.

3. Contribution as a reference-free evaluation metric is suspect and possibly subject to gaming: The authors point out that nearly all existing evaluation metrics (BLEU, METEOR, BLEURT, BERTScore) require reference text, which is characterized in the paper as a disadvantage. This is a bit of an odd argument to make for an *evaluation* metric, as this implies both the metric and the model are only given a source string to achieve their purposes. If the evaluation metric isn't given additional information (in the form of, e.g., a gold standard or reference), then on what basis is the metric supposed to do a better job at assessing the goodness of a target string, given that both the metric and the model are machine-learned using the same amount of input signal? If an evaluation metric is reference-free and proves indeed useful, then one can and probably should incorporate it directly inside the model or system (e.g., as a feature function or in a fine-tuning setup).  But incorporating the hallucination detection model as part of the generation model would render the former model rather useless as an *evaluation* metric. Note: I understand there is also a reference-based setup (which makes little difference), but the reference-free aspect is emphasized in the paper.

4.  No evaluation on any downstream task: But this could easily have been done as the model for hallucination detection does not require references: see point 3.

Overall, I think this paper's general direction with fine-tuning on hallucination classification data is probably worth pursuing. However, I think the paper's actual impact in its current form could be fairly limited, given the cons listed above. It is hard to draw conclusions from the main results of the paper (cons 1-2); the use as an evaluation metric is rather questionable (con 3); there is no application of the work on any end-to-end task (con 4).

Other comments:

Tables 2 and 4 are somewhat misleading in their references to different "models." Each of these tables evaluates hallucination classification with a *single* model, but the Models column refers to different models used to generate the underlying classification *data*. I think this could easily be misunderstood by the reader, as the established practice is that "models" refer to models for the task at hand (i.e., classification).

Experiments on hallucination detection are on MT and summarization, but the paper (and introduction in particular) discuss more generally "conditional neural sequence models" and mention other tasks such as free-form QA (Fan et al., 2019) and indirectly LM-based generation (Wang and Sennrich, 2020). By being reference-free, the work would, however, only apply to semantics-preserving tasks (output conveys and same meaning as the input or a subset) as summarization as MT. Otherwise, I don't see how to evaluate whether something is hallucinated without any reference.

Missed related work:

Improved Natural Language Generation via Loss Truncation. Daniel Kang, Tatsunori Hashimoto. ACL 2020.

Sticking to the Facts: Confident Decoding for Faithful Data-to-Text Generation. Ran Tian, Shashi Narayan, Thibault Sellam, Ankur P. Parikh.  arXiv 2019.


===========
Update: Thank you for your clarifications and updated paper, which addressed several of my concerns. I therefore increased my score by 1.

> First, we want to stress that we are not proposing a reference-free evaluation metric for quality estimation.

I understand it is not meant as a general quality estimation metric, and my point was more about the *reference-free* aspect. I tried to make a broader point which I think applies to any kind of reference-free metric (whether it is for quality estimation or specifically about hallucination).

---

> ### Author Response · Authors · 2020-11-18
> **Thank you for your feedback! We addressed your concerns.**
>
> Thank you and we really appreciate your valuable feedback! Please see our response below:
>
> > (1) Lack of experimental comparison and analysis
>
> We apologize that the presentation of the main results was confusing and lacked comparisons.
> Please see our general response (2) and (3) to this question. In our revised version, we added three proposed baseline methods for the new task and more complete analysis of the experimental results in Section 5.2. Compared with the baseline methods, we found that our model works well on both the cross-lingual task (MT) and monolingual task (summarization).
> In addition to that, we analyzed the distribution of part-of-speech (POS) tags of hallucinated words in Section 5.3 and found that our model predictions align well with that of the gold annotations.
> We also analyzed error rates on predicting synonyms as hallucinations in Appendix D.5.
>
> > (2) Results lack interpretability
>
> Thank you for pointing this out! We addressed your concern in our revised version by directly comparing our method against baselines in Tab.2 and moving the full results of precision, recall and F1 to Appendix D.3. In addition, we moved the comparisons of sentence-level hallucination assessment to Appendix D.2, as our focus is token-level hallucination prediction and these results are a little distracting to our reviewers.
>
> > (3) Contribution as a reference-free evaluation metric is suspect
>
> First, we want to stress that we are not proposing a reference-free evaluation metric for quality estimation. The existing evaluation metrics we pointed out are used for quality estimation, while we are focusing on hallucination detection and its potential applications in online generation systems where we can flag hallucinated words that may bring risks to users. For this purpose, the above mentioned metrics (BLEU, METEOR, BLEURT, BERTScore) cannot either perform token-level hallucination detection nor be used in the online generation system, as they require the reference sentences.
>
> Second, please read our general response (1) where we listed other potential ways of using these token-level hallucination labels.
>
> > (4) No applications on downstream tasks
>
> Please look at our general response (1), where we highlight two main benefits of this new task, in particular we demonstrate how to use token-level hallucination labels to improve low-resource machine translation in Section 6 of our revised paper and achieved significant improvements over strong baselines (including the loss truncation method as you pointed out in the missed reference) both in translation quality and reduction of hallucinations. We also list other potential ways of using these fine-grained labels to improve text generation in our general response (1)(b).
>
> > (5) Other comments: misleading table title “models”
>
> Thank you for pointing this out! We addressed your concerns in all the tables in our revised paper by removing “models” and directly comparing results on different test sets.
>
> > (6) Other comments: the proposed method only applies to semantics-preserving tasks
>
> We agree with you on this and we apologize that our mention of free-form question answering is misleading. In our revised version, we change that into “data-to-text generation”.

---

### Author Response · Authors · 2020-11-18
**General Responses to All Reviewers: [ADDED] we demonstrate how to use token-level hallucination with fine-grained loss in MT**

We really appreciate the valuable feedback and suggestions. In response to your reviews, we added a number of experiments and analysis to support our contributions, and we also improved the writing to clarify key points. We summarize them below:

>  (1) Resolve questions on the actual contributions/benefits of predicting hallucination labels at token-level:

We apologize that we didn’t explain the use cases of the proposed task well, and we want to stress that:

Predicting hallucination labels at token-level provides a tool for **diagnosing and interpreting model output**, which allows us to flag potential risks in online generation systems.
It also offers the possibility of using fine-grained token-level labels to **defend against hallucination and improve text generation**. To demonstrate this, in Section 6 of our revised paper, we use the hallucination labels with self-training (a semi-supervised method of data augmentation) in low-resource machine translation where the teacher model can generate hallucinated outputs for data from the low-resource domain. We use the predicted hallucination labels of outputs to introduce **a new fine-grained loss for target tokens**. This loss has a smaller penalty for errors on tokens that are likely hallucinations  (e.g. token-level loss truncation). We compare with strong baselines of noisy self-training (He et al, 2020) and sequence-level loss truncation (Kang & Hashimoto, 2020)) and find our methods significantly outperform them both in translation quality and reducing the amount of hallucination.

There are also many other potential ways to use token-level hallucination prediction to improve text generation, e.g. to "clean up" noisy parallel data in bi-text mining, acting as fine-grained rewards with reinforcement learning, or used as a ranking criterion at generation time, etc.
We have also highlighted the above points in the introduction of the revised paper.

> (2) Resolve question on the presentation of main results (Table 2. token-level predictions) and lack of baselines:

Thank you for pointing this out! We apologize that the presentation of this table was confusing and lacked comparisons. To improve this, we added three baseline methods (alignment based method, overlap-based method and synonym-based method). We also reformatted the table to make it more clear.

> (3) Adding analysis experiments:

We added a series of analysis experiments to provide more insights on the hallucination problem for the two tasks in the paper: (a) analysis on the Part-of-Speech tags and hallucination labels (please see Section. 5.3 and Appendix D.4 in our revised version for the macro and micro-level analysis, (b) analysis on the error rates of predicting synonyms as hallucinated words (Appendix D.5), (c) As a supplemental for Section. 6 (our proposed method of using hallucination labels), we provide examples of partially hallucinated outputs from a teacher and our model predictions in Appendix D.6.

> (4) Major revisions on writings:

Due to the space limit, (a) we merged the introduction and related work into one. (b) We put the sentence-level hallucination evaluation to the Appendix D.2 as this result is a little distracting to our reviewers. (c) We stressed in Section 2 that the definition of hallucination is only used as part of the guidelines for human annotators, who do not need to distinguish between the two different types of hallucinations.

We hope our responses could address your concerns and questions! We kindly request that you read these responses and let us know if you have any other questions.

---

### Decision · Program_Chairs · 2021-01-07
**Final Decision**

**Decision:**

Reject

**Comment:**


The paper proposes the novel task of detecting hallucinated tokens in sequence
generation, and a strategy to train such models using artificially generated
samples. The methods show reasonable correlation with human judgements.

The expert reviewers are unanimous in their lack of enthusiasm
about this work, with overall borderline assessments. The
reviewers provided some suggestions for improvement, and it is worth remarking
that the authors provided an impressive amount of work in the revised version,
addressing the suggestions. Specifically, they added baselines that validate that
the task is non-trivial, and the case study on improving machine translation.

In the discussion period, the reviewers appreciated the additions, and some
increased their rating, but the overall assessment remains borderline.  The
reviewers find the work lacks the expected amount of depth.  Some concerns
emphasized in the discussion period involve insufficient empirical analysis
(e.g., more NMT datasets and and analysis); understandable as this work was
added after submission, but still important.  A reviewer stresses concerns about
the definition of the task itself, which I agree is vague ("... cannot be
entailed by the sentence") and does not match the synthetic data generation
entirely, leading to unfortunate edge cases involving synonyms or -- worse --
slight narrowing that technically would still be entailment but maybe should be
considered unfaithful.  This casts doubt on the human evaluations and on
considering the task itself a main contribution, therefore leading to the
empirical framing that the reviewers perceive and expect.  It also seems to me
that there is a incremental, cat-and-mouse spirit to predicting automatically
generated hallucinations. In short, it seems like this paper is caught in
between trying to be a significant empirical contribution and a linguistically
well-motivated task and annotation project, and I understand that the reviewers
would prefer committing to one of these directions.

While I encourage the authors to pursue this direction more deeply,
in light of the borderline reviews, I do not recommend acceptance.